# Mediators of Metabolism: An Unconventional Role for NOD1 and NOD2

**DOI:** 10.3390/ijms22031156

**Published:** 2021-01-25

**Authors:** Megan T. Zangara, Isabel Johnston, Erin E. Johnson, Christine McDonald

**Affiliations:** 1Department of Inflammation & Immunity, Lerner Research Institute, Cleveland Clinic, Cleveland, OH 44195, USA; zangarm@ccf.org (M.T.Z.); johnsti@ccf.org (I.J.); exjohnson@jcu.edu (E.E.J.); 2Department of Molecular Medicine, Cleveland Clinic Lerner College of Medicine, Case Western Reserve University, Cleveland, OH 44106, USA; 3Department of Biology, John Carroll University, University Heights, OH 44118, USA

**Keywords:** NLR, metabolism, ER stress, mitochondria, hypoxia, high fat diet, metabolic syndrome, insulin resistance, obesity, diabetes

## Abstract

In addition to their classical roles as bacterial sensors, NOD1 and NOD2 have been implicated as mediators of metabolic disease. Increased expression of NOD1 and/or NOD2 has been reported in a range of human metabolic diseases, including obesity, diabetes, non-alcoholic fatty liver disease, and metabolic syndrome. Although NOD1 and NOD2 share intracellular signaling pathway components, they are differentially upregulated on a cellular level and have opposing impacts on metabolic disease development in mouse models. These NOD-like receptors may directly mediate signaling downstream of cell stressors, such as endoplasmic reticulum stress and calcium influx, or in response to metabolic signals, such as fatty acids and glucose. Other studies suggest that stimulation of NOD1 or NOD2 by their bacterial ligands can result in inflammation, altered insulin responses, increased reactive oxygen signaling, and mitochondrial dysfunction. The activating stimuli for NOD1 and NOD2 in the context of metabolic disease are controversial and may be a combination of both metabolic and circulating bacterial ligands. In this review, we will summarize the current knowledge of how NOD1 and NOD2 may mediate metabolism in health and disease, as well as highlight areas of future investigation.

## 1. Introduction

Nucleotide-binding, oligomerization domain (NOD) proteins NOD1 and NOD2 are intracellular pattern recognition receptors activated by peptidoglycan fragments [1,2]. Canonically, NOD1 recognizes γ-D-glutamyl-meso-diaminopimelic acid (iE-DAP), which is present primarily in Gram-negative bacteria, while NOD2 recognizes muramyl dipeptide (MDP) that is widely incorporated in the cell wall of almost all bacteria. After ligand binding, these NOD-like receptors (NLRs) homo-oligomerize and form a protein complex with receptor-interacting serine/threonine protein kinase 2 (RIP2). This complex then recruits and activates TGFβ activated kinase 1 (TAK1), which goes on to stimulate the inhibitor of nuclear factor ĸB (IĸB) kinase (IKK) complex and mitogen-activated protein kinase (MAPK) signaling cascades. This results in the activation of transcription factors, such as nuclear factor ĸB (NFκB), activator protein-1 (AP-1), and interferon regulator factor 5 (IRF5), and subsequently the expression of antimicrobial and pro-inflammatory molecules [1,2].

A well characterized feature of metabolic syndrome, diabetes, and obesity is low-grade, chronic inflammation characterized by elevated circulating levels of interleukin-1β (IL-1β), IL-6, IL-18, and tumor necrosis factor α (TNFα) [3]. It is thought that chronic inflammation may be a contributor to the development of metabolic diseases. NOD1 and NOD2 dysfunction are associated with other chronic inflammatory diseases, such as inflammatory bowel disease (IBD), asthma, arthritis, and periodontitis [1]. More recently, NOD1 and NOD2 have been implicated as mediators of human metabolic disease. Increased expression of NOD1 and/or NOD2 has been reported in a range of human metabolic diseases, including obesity, diabetes, non-alcoholic fatty liver disease (NAFLD), and metabolic syndrome [4,5,6,7,8,9], or chronic diseases associated with mitochondrial dysfunction, such as IBD [10]. These clinical associations have driven research investigating the role of these pattern recognition receptors in metabolic disease.

The unconventional metabolic roles for NOD1 and NOD2 can manifest through direct signaling, indirectly through shaping cellular stress responses, or by enhancing inflammation. Although several studies have begun to elucidate the molecular underpinnings of the metabolic roles of these pattern recognition receptors, these newly uncovered roles of NOD1 and NOD2 are not yet comprehensively defined. In this review, we will summarize the current knowledge of how NOD1 and NOD2 may mediate metabolism in health and disease, as well as highlight areas of future investigation.

## 2. Contribution of NOD1 and NOD2 to Metabolic Diseases

Metabolic disease can broadly be characterized as a spectrum of related diseases altering normal metabolism. While some are solely genetic, the majority of metabolic disease is driven by environmental factors (e.g., diet, activity level, etc.), sometimes in combination with genetic risk factors. The primary metabolic diseases in humans are diabetes and cardiovascular disease. However, the term “metabolic syndrome” is increasingly used to describe the co-occurrence of numerous metabolic disorders, including insulin resistance, obesity, hypertension, and dyslipidemia. Both NOD1 and NOD2 have been linked with innate immune system activation in metabolic disease, including insulin resistance, diabetes, and liver steatosis [4,5,6,7,8,9].

### 2.1. Links between Human Metabolic Diseases and NOD1 or NOD2

Clinical studies have provided insight into whether NOD1 or NOD2 signaling is potentially altered in human metabolic disease. A handful of studies have evaluated specific *NOD1* and *NOD2* polymorphisms to determine if they are associated with metabolic syndrome and insulin resistance. In one study, the carriage of non-synonymous polymorphisms of *NOD1* (rs2075820 [p.E266K]) and *NOD2* (rs2066842 [p.S268P]) was examined for links with metabolic syndrome and insulin resistance in 998 Canadians aged 20–29 [5]. Neither heterozygous nor homozygous carriage of these polymorphisms associated with metabolic syndrome biomarkers; however, individuals homozygous for the *NOD1* variant had positive correlations with increased saturated fat intake and insulin resistance that were not observed in other genotypes [5]. Molecular data from other studies suggest that saturated fatty acids can directly activate NOD1 pro-inflammatory signaling [11] and that the functional consequence of the E266K variant may be enhanced NOD1 activity [12]. Another study in 200 subjects of Turkish origin found no correlation between type 2 diabetes (T2D) or insulin resistance with carriage of a noncoding *NOD1* variant (rs5743336) or a frameshift mutation of *NOD2* (rs2066847 [c.3020insC]) previously linked to IBD risk [13]. Overall, there is not currently strong evidence of a genetic link of either *NOD1* or *NOD2* polymorphisms and the development of human metabolic disease.

In contrast, there are human studies demonstrating highly upregulated expression of NOD1 and NOD2 in individuals with metabolic syndromes. For example, a study of 25 Indian diabetic patients showed that diabetic monocytes had significant upregulation of both Nod1 (20 fold) and Nod2 (25 fold) transcripts that positively correlated with insulin resistance [8]. These monocytes had an activated phenotype, expressed higher levels of the NOD1/2 signaling components RIP2 and NFĸB, and produced higher levels of IL-6 and TNFα in comparison to monocytes from non-diabetic individuals. Additionally, monocytes from diabetic donors were more sensitive to stimulation with bacterial ligands for NOD1 or NOD2 (iE-DAP and MDP, respectively) than monocytes from non-diabetic individuals, potentially establishing an inflammatory feedback loop in these patients [8].

Other studies demonstrate a selective upregulation of NOD1 in adipose tissue during metabolic disease. Two studies, one comparing subcutaneous adipose tissue obtained from 18 women with gestational diabetes to those from 28 healthy pregnant patients [6] and the other examining subcutaneous adipose tissue taken from 34 subjects with metabolic syndrome [9], showed an increase of NOD1, but not NOD2 expression in individuals with metabolic disease. Both studies present similar findings illustrating that increased NOD1 expression in adipocytes results in heightened responsiveness to iE-DAP stimulation. This enhanced sensitivity to iE-DAP was linked to increased activation of NFĸB and elevated production of pro-inflammatory cytokines in patients with gestational diabetes or metabolic syndrome compared to those with normal glucose tolerance [6,9]. Taken together, these studies link aberrant NOD1 and NOD2 expression and activation, rather than genetic variants, with metabolic diseases.

### 2.2. High Fat Diet Consumption Upregulates NOD1 and NOD2 Signaling Pathways in Mice

The feeding of a high fat diet (HFD) to mice is used to mimic the consumption of a Western diet in humans that results in weight gain, increased adiposity, and a pro-inflammatory phenotype [14]. Observed also in mice fed a HFD are increased basal glucose and insulin levels, glucose intolerance, and insulin resistance. NOD1 and NOD2 expression are both upregulated in mice fed a HFD or a diet that mimics the high fat and high sugar content of a Western diet [15,16]. Nod1 transcript levels were more than doubled in adipose tissue of mice fed a HFD for 20 weeks and correlated with increased body weight and fasting blood glucose levels [16,17]. Nod2 expression was also amplified in distinct cell types after consumption of a HFD or Western diet (e.g., splenocytes and colonocytes) [15,18]. The amplification of NOD1 and NOD2 expression was associated with a higher basal inflammatory status detected by increased production of circulating IL-6, IL-1β, and TNFα. Upregulation of these NLRs appears to be dependent on circulating factors generated in response to HFD consumption, as this phenomenon not seen in genetically obese (*ob/ob*) mice [16] and in vitro stimulation of a HEK293T cell-based NOD1 reporter assay with serum from mice fed a HFD resulted in a significant increase in NOD1 activity [19]. The identity of the circulating factor is currently unknown and is somewhat controversial; however, there are several candidate mechanisms ranging from metabolic/dietary factors to microbial products that may be acting alone or in combination to upregulate NOD1 and NOD2 expression and signal transduction (Table 1). Regardless, it is clear that consumption of a HFD induces upregulation of both NOD1 and NOD2 expression in mice.

### 2.3. Opposing Roles of NOD1 and NOD2 in Metabolic Disease

One main aspect of the HFD diet murine model is induction of insulin resistance, a key hallmark of T2D, which is associated with chronic low-level inflammation. A role for NOD1 and NOD2 in HFD-induced insulin resistance was first demonstrated in *Nod1/2^−/−^* mice on a C57BL/6 background [7]. These double knockout mice are protected from many of the effects of a HFD, including inflammation, lipid accumulation, and peripheral insulin resistance; suggesting that together these two receptors play a key role in the development of metabolic disease caused by HFD and obesity (Figure 1) [7]. While this study only looked at the impact of a combined loss of NOD1 and NOD2, further studies have examined the requirement of these two receptors individually in the development of metabolic disease.

Similar to *Nod1/2^−/−^* mice, *Nod1^−/−^* mice were protected against development of metabolic syndrome when fed a HFD [38]. Surprisingly, this protection appears to be mediated by actions of NOD1 in both hematopoietic and non-hematopoietic cell compartments, as a hematopoietic-specific *Nod1* knockout mouse was only partially protected from HFD-induced insulin resistance [19]. Complementing these results are the findings that injection of the synthetic tetra-DAP NOD1 agonist FK156 into mice is sufficient to drive the onset of whole body insulin resistance [7] and exacerbate glucose intolerance in HFD fed mice in a manner that required RIP2 expression [22]. These findings suggest that NOD1 is an important mediator of diet-induced metabolic syndromes.

In stark contrast to the protective role of NOD1 loss in metabolic syndrome models, *Nod2^−/−^* mice fed a HFD had aggravated inflammation and obesity, as well as increased insulin resistance [39,40]. Complementary to these studies, BALB/c *Nod2^−/−^* mice have a loss of resistance to HFD-induced obesity [41]. When fed a HFD, BALB/c *Nod2^−/−^* mice became obese and developed numerous hallmarks of metabolic disease [41]. Additionally, chronic administration of the NOD2 ligand MDP either prior to the development of disease (preventatively) or after development of insulin resistance (therapeutically) resulted in improved insulin sensitivity and glucose tolerance in HFD fed wildtype mice [23]. Interestingly, while MDP treatment promoted these metabolic effects, it did not change overall body weight, adiposity, or serum endotoxin levels in these mice [23]. Further investigation uncovered that RIP2 expression in non-hematopoietic cells was required for the protective effects of MDP [22], as well as activation of the transcription factor IRF4 [22,23]. IRF4 is known to dampen adipose tissue and liver inflammation associated with obesity [23,42], as well as mediate anti-inflammatory effects of MDP treatment in the intestine [43]. IRF4 is not activated by NOD1 and may potentially explain the disparate actions of NOD1 and NOD2 activation in the HFD model [23]. The anti-inflammatory mechanisms of NOD2-stimulated IRF4 action are not completely defined; however, it appears that suppression of inflammation caused by metabolic endotoxemia may be one component, as HFD-fed C3H/HeJ mice with defective TLR4 responses were not responsive to MDP treatment [23]. Further investigation of the molecular mechanisms involved in key cell types (e.g., hematopoietic vs. stromal) and tissues (e.g., adipose, muscle, liver) is warranted to determine how activation of NLRs that converge on RIP2 results in opposing metabolic responses.

Additional evidence of a protective role for NOD2 in metabolic disease development is found in related mouse models of NAFLD. In this model, mice are fed a high fat, high fructose diet for 16 weeks that results in steatosis, fibrosis, and fat accumulation. Similar to the results on HFD-induced insulin resistance, both whole body and hepatocyte-specific *Nod2* knockout mice exhibited higher levels of liver steatosis and fibrosis when fed a NAFLD-inducing diet [4]. These changes were associated with alterations in the metabolic transcriptome of the liver in *Nod2^−/−^* mice, suggesting that NOD2 plays a critical role in shaping the metabolic, inflammatory, and fibrotic function of the liver in response to dietary cues. Taken together, these studies demonstrate that NOD1 activation may promote metabolic disease development, while the actions of NOD2 protect against the development of diet-induced metabolic disease in mouse models.

## 3. NOD1 and NOD2 as Modulators of Cellular Responses to Glucose and Fatty Acids

One mechanism proposed to drive activation of NOD1 or NOD2 in metabolic disease is excess nutrients generated during overfeeding conditions acting as endogenous ligands or danger-associated molecular patterns (DAMPs) to stimulate inflammation. Two potential nutrients that may act as novel NOD1/2 stimulatory ligands are glucose and fatty acids [11,16,28,29]. Alternatively, obesity, diabetes, and hyperglycemic states have been shown to cause intestinal microbiome dysbiosis and increased intestinal permeability that can lead to an increase in circulating microbial products [44,45,46]. These circulating microbial ligands can trigger canonical activation of NOD1 and NOD2 to alter other metabolic signals within cells and tissues required to maintain overall metabolic homeostasis, such as glucose uptake or insulin sensitivity (Table 1).

### 3.1. Roles of NOD1 and NOD2 in Hyperglycemia and Insulin Resistance

A central feature of many metabolic diseases is insulin resistance driven by hyperglycemia. A small number of studies indicate that NOD1 and NOD2 may mediate hyperglycemic responses. For example, high levels of glucose have been demonstrated to activate NOD1 signaling in mesangial cells [28]. Glucose may also activate NOD2 during hyperglycemia-induced podocyte dysfunction observed in diabetic nephropathy [29]. Although more studies need to be performed examining the direct effects of glucose on NOD1 and NOD2 activation, these studies suggest that glucose can act as a novel activating ligand of NOD1 and NOD2.

Skeletal muscle cells play a vital role in whole body insulin resistance as they are a predominant site for insulin-controlled glucose uptake. In muscle cells, activation of NOD2 alone was able to acutely induce cell autonomous insulin resistance; when the NOD2 ligand MDP was added to myotubes in vitro, insulin resistance rapidly arose within 3 h [24]. MDP-stimulated insulin resistance was associated with increased inhibitory serine phosphorylation of insulin receptor substrate-1 (IRS-1) and reduced activating IRS-1 tyrosine phosphorylation, as well as decreased GLUT4 translocation and insulin-stimulated glucose uptake. These insulin signaling alterations were accompanied by increased pro-inflammatory signaling mediated by MAPKs and NFĸB. Treatment of myotubes with NOD1 activating ligands did not alter insulin-stimulated glucose uptake, indicating divergent roles for NOD1 and NOD2 on glucose regulation in skeletal muscle cells [24]. These findings contrast with the results from the HFD mouse models that demonstrate increased glucose sensitivity after treatment with MDP [23]. Although the mechanisms underlying this apparent discrepancy are not fully understood, it suggests a more complex situation in vivo that may include compensatory metabolic mechanisms (e.g., upregulation of insulin levels in response to skeletal muscle glucose resistance), crosstalk between skeletal muscle and other metabolically active tissues, such as adipose or liver, alterations in inflammatory cell types, or involvement of microbiota signals, which determine the overall metabolic health of an organism [7,22,23,47].

Another major site of glucose uptake regulated by insulin levels is adipose tissue. Although both NOD1 and NOD2 are expressed in adipose tissue, only NOD1 expression is elevated in obese patients, individuals with metabolic syndrome, or mice fed a HFD [6,9,16,17]. In vitro stimulation of human or mouse adipocytes with the NOD1 ligand iE-DAP increased pro-inflammatory responses and weakened insulin receptor signaling, resulting in reduced insulin-stimulated glucose uptake and insulin resistance [16,17]. These metabolic alterations were demonstrated to be mediated by NOD1, as knockdown of NOD1 expression restored insulin receptor signaling in iE-DAP stimulated 3T3-L1 mouse adipocytes [16]. These findings demonstrate a key role for NOD1 in adipocytes and adipose tissue in regulating signaling pathways that shape insulin resistance.

### 3.2. Fatty Acids as Ligands for NOD1 and NOD2

Diets high in saturated fat are drivers of metabolic syndrome, cardiovascular disease, and diabetes [48]. Conversely, several studies indicate that dietary unsaturated fatty acids can be protective against these metabolic diseases. Acute activation of NOD1 in adipocytes leads to lipolysis via activation of NFκB and protein kinase A (PKA) signaling [49,50]. Lipolysis results in an increase in circulating fatty acids, which in turn can lead to insulin resistance, and so is likely to be in part responsible for metabolic phenotypes seen in vivo [51]. Multiple molecular and cellular mechanisms appear to mediate the metabolic effects of fatty acids, ranging from alterations in cell membrane fluidity to stimulation of nuclear hormone receptors. Studies indicate that saturated fatty acids, such as lauric acid, stimulate activation of NOD1 and NOD2 to induce NFĸB activation and pro-inflammatory cytokine production from intestinal epithelial cells and adipocytes [11,30]. In adipocytes, it was also demonstrated that saturated fatty acids impaired insulin-stimulated glucose uptake and this was partially reversed by NOD1 knockdown [11]. Conversely, unsaturated fatty acids, such as docosahexaenoic acid (DHA), inhibit bacterial ligand induced inflammation by impairing NOD1 and NOD2 self-oligomerization. Interestingly, DHA was demonstrated to also downregulate saturated fatty acid-stimulated NOD1 and NOD2 signaling, indicating a dominant negative effect of unsaturated fatty acids on inflammation. These studies demonstrate that dietary fatty acids modulate NOD1 and NOD2 activity and may impact the development of metabolic disease.

## 4. NOD1 and NOD2 as Mediators of Cellular Metabolic Stress

The activating stimuli for NOD1 and NOD2 in the context of metabolic disease is controversial and may be a combination of metabolic and bacterial ligands, as well as involve direct and indirect mechanisms of action. While NOD1 and NOD2 have been demonstrated to alter systemic metabolic phenotypes (e.g., obesity, insulin sensitivity, etc.), there is also evidence that they play a role in cellular metabolic stress signaling. For example, HFD-induced obesity has been shown to be a strong activator of the unfolded protein response (UPR) in mouse liver, which can be ameliorated through the NOD1 and NOD2 dampening actions of troxerutin, a derivative of natural bioflavonoid rutin found in tea, coffee, and many fruits and vegetables. Troxerutin alleviates HFD-induced obesity parameters, as well as diminishes the UPR upregulated expression of pro-inflammatory cytokines in the liver [36]. Troxerutin imparts this effect to some degree by reducing NOD1 and NOD2 expression, as well as decreasing NOD1 and NOD2 signaling by disrupting functional interactions of these pattern recognition receptors with the kinase RIP2 [36].

### 4.1. Endoplasmic Reticulum Stress

Endoplasmic reticulum (ER) stress, which is caused by accumulation of unfolded or misfolded proteins, has been implicated in the pathogenesis of diabetes, heart disease, and nutrient deprivation [52]. The UPR has three main stress sensors: protein kinase RNA-like ER kinase (PERK), inositol-requiring enzyme 1 (IRE1), and activating transcription factor 6 (ATF6) [52]. Activation of PERK constitutes the immediate response to ER stress, which in turn phosphorylates eukaryotic translation initiator factor 2α (eIF2α). This halts general protein translation, activates nuclear erythroid 2-related factor 2 to aid in redox metabolism, and allows ATF4 mRNA translation in order to upregulate transcription of genes involved in amino acid metabolism, apoptosis, autophagy, and antioxidant responses [53]. Maintained or additional ER stress leads to the activation of the remaining two sensors. IRE1α activation results in dimerization and autotransphosphorylation, allowing it to excise a 26-nucleotide-long intron in the mRNA of X box-binding protein 1 (XBP1). Spliced XBP1 (XBP1s) is a transcription factor that will upregulate the transcription of genes involved in ER-associated degradation (ERAD), protein translocation to the ER, and protein folding [53]. Lastly, ATF6 will translocate to the Golgi under ER stress conditions to be processed by site-1 and site-2 proteases, releasing a cytosolic fragment (ATF6f). ATF6f directly regulates the genes encoding XBP1 and ERAD components [53].

Recently, NOD1 and NOD2 have been shown to directly link ER stress and inflammation in multiple cell types, although the precise mechanisms underlying this linkage are not completely understood. Activation of IRE1α normally leads to splicing of XBP1, but under certain “alarm stress” signals it can interact with TNF receptor-associated factor 2 (TRAF2) to activate NFκB [53]. One study demonstrated that activation of this UPR pathway by thapsigargin and subsequent IL-6 production from mouse bone-marrow derived macrophages (BMDMs) required both NOD1 and NOD2 expression, as well as the downstream kinase RIP2 [31]. The UPR activated by HFD-induced obesity is associated with activation of the IRE1α and PERK pathways, as demonstrated by increased phosphorylation of PERK, eIF2α, and IRE1α [52]. TRAF2, which is downstream of IRE1α, is also upregulated, leading to increased expression of the NFκB target genes, IL-1β, TNFα, and MCP-1. Similar findings were reported when ER stress was induced in response to dithiotheritol, *Brucella abortus* VceC effector protein, or *Chlamydia muridarum* infection. ER stress induced IL-6 secretion is blocked by tauroursodeoxycholic acid (TUDCA), a bile acid that promotes protein folding and reduces ER stress [37]. Further data indicates that activation of NOD1 and NOD2 by ER stress in BMDMs is a distinct pathway from canonical NOD1/2 signaling, as TUDCA treatment did not decrease production of IL-6 in response to either iE-DAP or MDP [31]. *In vivo*, similar results were observed using both chemical and bacterial inducers of ER stress. Thapsigargin injection resulted in elevated serum levels of the pro-inflammatory cytokines IL-1, KC/CXCL1, and MIP-1β [31]. This inflammatory response did not occur in *Nod1/2^−/−^* mice injected with thapsigargin, or when thapsigargin injected wildtype mice were treated with TUDCA. Likewise, pregnant mice infected with *B. abortus* were protected from placental inflammation and had improved pup survival when treated with TUDCA or were deficient in NOD1/2 expression.

Different findings were reported in studies examining human monocyte-derived macrophages (MDM) [32,33]. One study demonstrates by RNAi-mediated knockdown that thapsigargin-induced ER stress does not require expression of NOD1 or NOD2, either alone or in combination [32]. Instead, their data revealed that MDP induces an UPR that involves all three branches of the ER stress response mediated by interaction of NOD2, RIP2, and laccase domain containing-1 (LACC1) at the ER membrane [32]. Another study indicates that a bacterial ligand able to stimulate both NOD1 and NOD2 (M-triDAP) does not induce ER stress when added to MDM alone [33]. Instead, M-triDAP enhanced ER stress induced by 2-deoxy-D-glucose (2-DG) that results in increased pro-inflammatory cytokine production via IRE1α mediated activation of XBP1s and p38 MAPK [33]. Taken together, these studies reveal new roles for NOD1 and NOD2 as mediators of ER stress-induced inflammation in macrophages.

The role of NOD1 and NOD2 in directly mediating ER stress signals also appears to be cell type dependent. In astrocytes, parkin (encoded by *Park2*) has been shown to mediate the response to ER stress inducing signals [54]. Genetic deletion of parkin increases NOD2, but not NOD1 expression, and enhances UPR basally, as evidenced by increased expression of the ER stress regulated genes: XBP1s, ATF6, ATF4, CHOP, and Ccl2 in primary astrocytes [54]. Transient knockdown of NOD2 expression in parkin-deficient astrocytes was sufficient to reverse this exaggerated UPR. Interestingly, parkin not only influences basal NOD2 expression, but also regulates its function by targeting NOD2 for ubiquitylation and subsequent proteasome degradation [54]. This study suggests that the requirement of NOD1 and NOD2 for mediating ER stress responses may be cell type dependent; in BMDMs both NOD1 and NOD2 are required [31], while NOD2 plays a dominant role in astrocytes [54].

In addition to direct mediation of an IRE1α triggered ER stress response, NOD2 plays a second role in shaping the cellular response to ER stress by influencing the expression levels of UPR mediators. Tunicamycin (TM) inhibits protein glycosylation and blocks protein transit from the ER to the Golgi apparatus, inducing ER stress. In vascular smooth muscle cells (VSMCs), ER stress-induced cell death via TM treatment was significantly increased with loss of NOD2 expression, and correlated with an approximately 60% reduction of IRE1α and eIF2α expression [35]. Interestingly, expression of other mediators of the UPR are either amplified (e.g., PERK) or remain unchanged (e.g., ATF4) with loss of NOD2 expression [35]. Overexpression of NOD2 in VSMCs treated with TM showed increased levels of IRE1α, suggesting that NOD2 regulates IRE1α expression under ER stress conditions [35]. The blunting of the IRE1α-mediated ER stress response in NOD2-deficient VSMCs is mediated by a reduction of XBP1s protein levels, and modest reductions in Grp78, Pdi-1, and Herpud1 transcript levels [35]. Reconstitution of NOD2-deficient VSMCs with a NOD2 expression construct reverses the dampened IRE1α signaling response [35]. These results suggest that NOD2 is required for proper signaling of the IRE1α signaling arm of the unfolded protein response.

Finally, NOD1 has uniquely been demonstrated to mediate ER stress signaling through a second arm of the UPR characterized by PERK activation [21,55]. In a study focused on factors that make cells more susceptible to *Salmonella enterica* serovar Typhimurium infection, it was found that ER stress inducers that engage PERK made cells more responsive to infection in a NOD1-dependent manner [21]. Thapsigargin treated cells had a higher level of NFĸB activation and transcription of pro-inflammatory genes (e.g., IL-6 and IL-23) in response to iE-DAP, which was blocked by the PERK inhibitor GSK2656157, but not inhibitors of IRE1α (KIRA6) or ATF6 (4(-2-aminoethyl) benzenesulfonyl fluoride hydrochloride) [21]. Complementary to these findings are data supporting a role for NOD1 in ER stress activated cell death. Cell death in an oxygen-glucose deprivation and reperfusion (OGD/R) in vitro model of cerebral ischemia-reperfusion was shown to be partially mediated by induction of ER stress signaling pathways [55]. OGD/R treatment of primary rat cortical neurons induces NOD1-dependent increases in CHOP, cleaved caspase-12, and cleaved caspase-3. The activation of this PERK pathway results in decreased cell viability, increased lactate dehydrogenase release, and ultimately enhanced cell apoptosis. Knockdown of NOD1 levels can reverse the upregulation of ER stress pathways and decrease cell apoptosis [55]. These studies reveal a critical role for NOD1 in mediating PERK pathway induced inflammation and ischemia-reperfusion injury, further supporting the concept that NOD1 is a signaling intermediate for ER stress responses.

### 4.2. Calcium Signaling

NOD1 and NOD2 may also be activated in response to calcium influx, as thapsigargin induces ER stress through inhibition of the sarco endoplasmic reticulum Ca^2+^ ATPase (SERCA) that results in calcium influx. In intestinal epithelial cell lines and murine derived intestinal organoids, thapsigargin treatment results in a NOD1, NOD2, and RIP2 dependent transcriptional increase in pro-inflammatory cytokines CXCL1 and IL-8, and the chemokine CCL20 [34]. Similarly, NOD1 and NOD2-dependent cytokine/chemokine induction was also induced in response to the calcium ionophore A23187, but not other ER stress inducers like TM or the SubAB toxin of Shiga toxigenic *Escherichia coli* in intestinal epithelial cells [34].

It is unclear, however, whether NOD1 and NOD2 are activated directly by calcium signaling or through calcium-dependent influx of trace peptidoglycan fragments present in serum [34]. Using high pressure liquid chromatography coupled mass spectrometry, it was revealed that trace amounts of peptidoglycan fragments are detectable in animal serum. The NOD1 and NOD2-dependent increases in IL-8 production from thapsigargin treated intestinal epithelial cells were found to be serum dependent and blocked by endocytosis inhibition [34]. These findings indicate that NOD1/2-dependent induction of inflammation can be a combination of direct, non-canonical activation by ER stress, as well as potential stimulation of canonical NOD1/2 signaling by trace peptidoglycan fragments internalized by concomitant calcium flux during ER stress [34].

### 4.3. Reactive Oxygen Species Generation and Mitochondrial Dysfunction

Mitochondria are important metabolic mediators, as sites of the citric acid cycle which derives energy from carbohydrates, fats, and proteins for cellular use. States of nutrient excess can cause excess reactive oxygen species (ROS) production, leading to mitochondrial damage and further energy imbalances [56]. Evidence suggests that NOD1 and NOD2 play both positive and negative roles in ROS generation and subsequent mitochondrial dysfunction that may mediate the pathogenesis of metabolic disease.

NOD1 is one of the most highly expressed pattern recognition receptors in brown adipose tissue (BAT), whereas NOD2 is barely detectable [20]. In the context of obesity, only NOD1 expression is further upregulated in BAT from mice fed a HFD or in genetically obese (*ob/ob*) mice. BAT produces heat via oxidative phosphorylation uncoupling from ATP synthesis. Activation of NOD1 by iE-DAP in BAT suppressed expression of an important mediator of this process, uncoupling protein 1 (UCP-1), via suppressing UCP-1 promoter activity [20]. This resulted in a decreased mitochondrial basal oxygen consumption rate and increased basal extracellular acidification rate, which approximates glycolysis and lactate production [20]. Additionally, isoproterenol-induced uncoupling of the oxygen consumption rate from proton leakage was suppressed in iE-DAP stimulated cells, suggesting that NOD1 activation promotes BAT dysfunction by impairing mitochondrial function [20].

In contrast to the results from BAT, NOD2 plays a more dominant role in ROS generation and altered mitochondrial function in skeletal muscle and dendritic cells. In rat L6 skeletal muscle cells and human monocyte-derived dendritic cells (MDDC), mitochondrial reactive oxygen species (mtROS) levels increased in a dose-dependent manner after NOD2 stimulation with MDP [25,27]. Treatment with the antioxidant n-acetyl cysteine (NAC), the NADPH oxidase inhibitor diphenyleneiodonium chloride (DPI), or the mitochondrial respiratory chain inhibitor rotenone quenched this response [25]. Additionally, MDP increased the amount of protein carbonyls (a result of oxidative stress), enhanced glutathione peroxidase activity, and stimulated hydrogen peroxide release [25]. In skeletal muscle cells, MDP treatment also induced mitochondrial dysfunction, evidenced by mitochondrial depolarization, decreased citrate synthase activity, reduction of total cellular ATP levels, and decreased maximal respiration [25]. Again, treatment with NAC resolved this mitochondrial dysfunction, as well as MDP-induced pro-inflammatory signaling mediated by activation of the MAPKs, JNK, ERK, and p38. Conversely, MDP treatment of MDDC resulted in enhanced mitochondrial respiration, increased mtROS, and improved antimicrobial function [27]. These results suggest that canonical NOD2 signaling can positively impact the antimicrobial function of MDDC by increasing mitochondrial function and mtROS production [27], while also negatively impacting metabolic function of skeletal muscle cells through induction of mitochondrial dysfunction driving mtROS production [25].

In contrast, NOD2 has a positive role in preventing mtROS production and mitigating mitochondrial dysfunction in intestinal stem cells (ISCs) [26]. Radiation therapy is effective in treating many types of cancer because it disproportionately targets rapidly dividing cells; however, this often results in enteritis due to increased apoptosis of rapidly dividing ISCs. Shortly after irradiation, ISCs initiate apoptosis, and this cell death is mainly driven by the high levels of ROS created via water radiolysis [26]. In vitro studies demonstrated that two grays of radiation induced high levels of total and mtROS in murine small intestine-derived organoids, with an average 50% lethality. Pre-treatment of organoids with the NOD2 ligand MDP mitigated mtROS production and increased survival by 33% through induction of autophagy to remove radiation-damaged mitochondria (mitophagy), rather than a pro-inflammatory response in ISCs [26]. Overall, these studies highlight the cell type specific responses of NOD1 and NOD2 that modulate ROS production and mitochondrial dysfunction that may contribute to metabolic disease.

## 5. Discussion

Metabolic diseases affect a large proportion of the global population and are on the rise. In a global survey of 195 countries, it was found that 604 million adults and 108 million children were obese in 2015 [57]. These results reflect a tripling of prevalence from estimates in 1980 and these numbers are anticipated to continue to increase. In the United States, the total economic cost of obesity and chronic diseases associated with obesity is estimated to be USD 1.72 trillion [58]. More recently, established metabolic disease has been identified as an independent risk factor for severe COVID-19 and, conversely, infection with SARS-CoV-2 can increase vascular complications associated with diabetes and obesity [59]. Understanding the nuances of the roles NOD1 and NOD2 play in metabolism may be essential for effective therapies for these prevalent metabolic diseases, chronic inflammatory diseases, or infection.

Further research needs to be performed to fully understand the mechanisms by which NOD1 and NOD2 contribute to the modulation of metabolic responses in different cell types and tissues. Although precise mechanisms remain somewhat controversial, it appears that these NLRs can induce metabolic stress in response to their canonical bacterial ligands, as well as propagate stress signals generated from the ER, mitochondria, or calcium flux. Additional studies are needed to dissect the contribution of these often intersecting stress signals to the activation of NOD1 and NOD2 induced inflammation, keeping in mind that these signals may be distinct in different cell types or tissues.

Similarly, the nature of the NOD1 and NOD2 activating signals in the context of metabolic disease is yet to be conclusively defined with studies indicating the existence of circulating factors, which could be dietary or microbial (or a combination of both) that activate these NLRs. Studies implicate the intestinal microbiota as a source of these circulating factors that leak from the gut to the circulation through an impaired intestinal barrier associated with many metabolic diseases [4,34,38,40,60]. However, it is unclear whether these microbiome-derived NOD1/2 stimulatory factors are canonical peptidoglycan ligands or microbial metabolites. The results of these types of studies could provide novel microbiome-directed therapies to control inflammation and metabolism.

Additionally, a major gap in our understanding is the opposing roles NOD1 and NOD2 play in the control of diet-induced obesity and insulin resistance. Although these NLRs share common signaling mediators to respond to bacterial infection and HFD consumption, the disparate roles of NOD1 and NOD2 in HFD mouse models suggest the use of novel and distinct signaling pathways by NOD1 and NOD2 downstream of RIP2 in the context of metabolic stress [22]. One proposed mediator of these opposing roles is the engagement of IRF4 by NOD2 to induce a protective response [23]; however, it is not understood how NOD2 activates this transcription factor or the molecular switches that control the induction of this response by NOD2 and not the closely related NOD1.

Finally, many of the proposed roles for NOD1 and NOD2 in metabolic disease have been defined in animal models and require more extensive investigation in larger studies of human metabolic disease. Functional investigation of these NLRs from patient-derived tissues are critical to complement the associations of increased NOD1 and/or NOD2 expression with disease. Although genetic variants in *NOD1* and *NOD2* have not been identified in genome wide association studies (GWAS) of diabetes or other metabolic syndromes, this may not be surprising due to the strong influence that environmental factors have in disease development. Additionally, the majority of GWAS have been performed on Caucasian populations, so it may be beneficial to perform additional GWAS in other ethnic populations with higher risk for metabolic disease, such as Native Americans, Southeast Asians, or African Americans, to determine whether *NOD1* and *NOD2* variants are genetic risk factors for metabolic dysfunction.

## Figures and Tables

**Figure 1 ijms-22-01156-f001:**
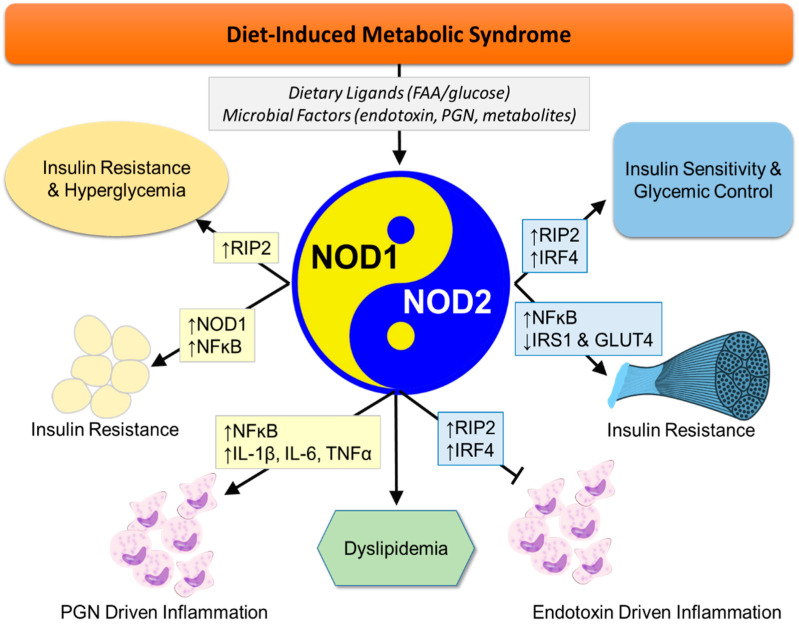
NOD1 and NOD2 function in different cell types and tissues to result in opposing roles in diet-induced metabolic disease. Peptidoglycan (PGN) activation of NOD1 promotes insulin resistance and hyperglycemia, while activation of NOD2 restores insulin sensitivity and glycemic control. Shown in boxes are proposed molecular mediators, with NOD1-dependent roles shaded in yellow, NOD2-dependent roles shaded in blue, combined actions shaded in green.

**Table 1 ijms-22-01156-t001:** Metabolic Actions of NOD1 and NOD2 Ligands.

Ligand	Target	NOD1 Action	NOD2 Action
***Microbial***			
iE-DAP	Adipocytes [16,17,20]	↑ pro-inflammatory signaling ↑ insulin resistance ↓ glycoysis ↓ OxPhos uncoupling	
	Epithelial Cells [21]	↑ pro-inflammatory signaling	
FK156	Mice on HFD [7,22]	↑ insulin resistance ↑ glucose intolerance	
MDP	Mice on HFD [22,23]		↑ insulin sensitivity, ↑ glucose tolerance
	Skeletal Muscle Cells [24,25]		↑ insulin resistance ↓ IRS-1 activation, ↓ glucose uptake ↑ mtROS
	Intestinal Stem Cells [26]		↓ mtROS
	Dendritic Cells [27]		↑ mtROS and mitochondrial respiration
***Dietary***			
Glucose	Mesangial Cells [28]	↑ pro-inflammatory signaling	
	Podocytes [29]		↑ pro-inflammatory signaling
Saturated Fatty Acids	Adipocytes [11] Fibroblasts [12] Intestinal Epithelial Cells [30]	↑ pro-inflammatory signaling ↓ insulin-stimulated glucose uptake	↑ pro-inflammatory signaling
Unsaturated Fatty Acids	Intestinal Epithelial Cells [30]	↓ pro-inflammatory signaling	↓ pro-inflammatory signaling ↓ NOD2 oligomerization
***Endoplasmic Stress Agonists***			
Thapsigargin	Mice [31]	↑ IRE1α activation ↑ IL-1β, IL-6, TNFα, MCP-1	↑ IRE1α activation ↑ IL-1β, IL-6, TNFα, MCP-1
	Macrophages * [31,32,33]	↑ IRE1α activation * ↑ IL-1β, IL-6, TNFα, MCP-1	↑ IRE1α activation * ↑ IL-1β, IL-6, TNFα, MCP-1
	Intestinal Epithelial Cells [34]	↑ CXCL1, IL-8, CCL20	↑ CXCL1, IL-8, CCL20
Tunicamycin	VSMC [35]		↑ IRE1α activation ↓ cell death
***Endoplasmic Stress Antagonists***			
Troxerutin	Mice on HFD [36]	↓ NOD1 expression ↓ pro-inflammatory signaling	↓ NOD2 expression ↓ pro-inflammatory signaling
TUDCA	Mice [31,37]	↓ IRE1α activation	↓ IRE1α activation
	BMDM [31,37]	↓ IRE1α activation	↓ IRE1α activation
PERK Inhibitor GSK2656157	Epithelial Cells [21]	↓ pro-inflammatory signaling	
***Calcium Flux Agonists***			
A23187	Intestinal Epithelial Cells [34]	↑ CXCL1, IL-8, CCL20	↑ CXCL1, IL-8, CCL20

↑ = increased levels; ↓ = decreased levels; * conflicting mechanisms reported.

## Data Availability

Not applicable.

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
