# Peer review of "Mediators of Metabolism: An Unconventional Role for NOD1 and NOD2"

_ijms, 2021, doi:10.3390/ijms22031156_

Round 1
Reviewer 1 Report
McDonald and co-works present a very well written comprehensive and timely review article on the role of NOD1 and NOD2 in metabolic disorders.
Only recently several labs reported on a novel function of NOD1 and NOD2 in obesity and the authors provide a throughout discussion of these data and propose a functional framework of the functions of NOD1 and NOD2.
The article very well fits the topic of the special issue and is of interest for a broader readership.
I only have some minor comments that should be addressed:
- Line 43: “It is well established that chronic inflammation is a key factor in the development of metabolic disease”. This reviewer respectfully disagrees with this notion. It is well established that inflammation is associated with obesity, however so far functional data that would allow to establish a causality, as proposed by the authors is lacking. This statement should be redefined.
- Line133: T2D should be defined.
- Lines 210 ff.: This is contrast to the data from NOD2 ko mice discussed above. Could the authors provide an explanation or at least comment on this?
Author Response
We thank the reviewer for taking the time to provide a detailed and constructive review of our manuscript. We have addressed your comments in the manuscript as follows:
-
Line 43: “It is well established that chronic inflammation is a key factor in the development of metabolic disease”. This reviewer respectfully disagrees with this notion. It is well established that inflammation is associated with obesity, however so far functional data that would allow to establish a causality, as proposed by the authors is lacking. This statement should be redefined. We agree that causality between inflammation and obesity has not conclusively been demonstrated; therefore, we have reworded the sentence to: "It is thought that chronic inflammation may be a contributor to the development of metabolic disease."
-
Line133: T2D should be defined. This abbreviation is defined on page 2 in the following sentence: "Another study in 200 subjects of Turkish origin found no correlation between type 2 diabetes (T2D) or insulin resistance..."
-
Lines 210 ff.: This is contrast to the data from NOD2 ko mice discussed above. Could the authors provide an explanation or at least comment on this? I believe the reviewer is referring to the contrasting data observed for NOD2 in skeletal muscle cells and whole animal studies. Added to Section 3.1, second paragraph, page 5 is the following comment: "These findings contrast with the results from the HFD mouse models that demonstrate increased glucose sensitivity after treatment with MDP [25]. Although the mechanisms underlying this apparent discrepancy are not fully understood, it suggests a more complex situation in vivo that may include compensatory metabolic mechanisms (e.g. upregulation of insulin levels in response to skeletal muscle glucose resistance), crosstalk between skeletal muscle and other metabolically active tissues, such as adipose or liver, alterations in inflammatory cell types, or involvement of microbiota signals, which determine the overall metabolic health of an organism.”
Reviewer 2 Report
The authors have written a timely and comprehensive review on the role of Nod1 and Nod2 in metabolism. They have presented the data and conclusions in a clear and concise manner. They have highlighted the many gaps in our knowledge, which requires more extensive investigation in this area of research. Also, they have described the opposing roles of Nod1 and Nod2, which is an intriguing aspect of these PRRs.
There are a few minor suggestions:
There are some recent papers on Nod 1/2 and metabolism that the authors should consider including:
J. Biol. Chem. (2020). Glycolytic reprogramming of macrophages activated by NOD1 and TLR4 agonists: No association with proinflammatory cytokine production in normoxia.
Front. Immunol. (2019). Ataxin-3 links NOD2 and TLR2 mediated innate immune sensing and metabolism in myeloid cells.
Cell Death & Disease (2020). NOD2 inhibits tumorigenesis and increases chemosensitivity of hepatocellular carcinoma by targeting AMPK pathway.
Cell Rep. (2019). LACC1 required for NOD2-induced, ER stress-mediated innate immune outcomes in human macrophages and LACC1 risk variants modulate these outcomes.
Fig. 1: Dyslipidemia is observed with both Nod1 and Nod2 and should be included in the figure.
Reference 11 (Chin Med Sci J 2013, 28, 211-217, 508 doi:10.1016/s1001-9294(14)60004-3), cannot be accessed through PubMed or other major databases and thus should not be included.
Please provide a reference(s) for the lines 233-234, “Acute activation of NOD1 in adipocytes leads to lipolysis via activation of NFκB and protein kinase A (PKA) signaling.”
Lines 239-241: The authors state that saturated fatty acids directly activate Nod1 and Nod2. However, the paper cited (ref. 36) does not show direct activation. The majority of experiments in the cited paper are done by stimulating cells for 18 h and likely activation of inflammatory pathways is an indirect effect.
Lines 248-249: The authors state that, “…dietary fatty acids directly modulate NOD1 and NOD2…”. The experiments in the cited paper are all done in vitro and no conclusion can be made about dietary fatty acids and as mentioned in the previous comment the data does not demonstrate direct activation of these PRRs.
Section 4.1 is long compared to other sections and could be shortened.
Lines 324-325: The authors state that, “…expression of IRE1α and eIF2α was also drastically reduced”. The expression of eIF2α is significantly reduced but only by ~60%, which is not a drastic reduction.
Author Response
We thank the reviewer for taking the time to provide a detailed and constructive review of our manuscript. We have addressed your comments in the manuscript as follows:
- There are some recent papers on Nod 1/2 and metabolism that the authors should consider including. We thank the reviewer for drawing our attention to these additional papers related to the topic of the review manuscript. We have incorporated information from 3 of the 4 recommended papers into the revised manuscript in Section 4.1 and Section 4.3. We did not include the paper by Ma et al. (2020) Cell Death & Disease, as we felt it was more focused on cell death mechanisms in tumorgenesis than metabolic disease.
-
Fig. 1: Dyslipidemia is observed with both Nod1 and Nod2 and should be included in the figure. This has been added to the figure.
-
Reference 11 (Chin Med Sci J 2013, 28, 211-217, 508 doi:10.1016/s1001-9294(14)60004-3), cannot be accessed through PubMed or other major databases and thus should not be included. We respectfully disagree with the reviewer and will keep the publication in the manuscript. This article is indexed in PubMed and includes a doi link that directs the reader to Science Direct to obtain access to the full text. Although this is not an open access article or a PMC article, it is still widely available to researchers via institutional subscriptions to Science Direct or Iliad library request.
-
Please provide a reference(s) for the lines 233-234, “Acute activation of NOD1 in adipocytes leads to lipolysis via activation of NFκB and protein kinase A (PKA) signaling.” Thank you for catching this oversight; the two applicable references have now been included.
-
Lines 239-241: The authors state that saturated fatty acids directly activate Nod1 and Nod2. However, the paper cited (ref. 36) does not show direct activation. The majority of experiments in the cited paper are done by stimulating cells for 18 h and likely activation of inflammatory pathways is an indirect effect.
Lines 248-249: The authors state that, “…dietary fatty acids directly modulate NOD1 and NOD2…”. The experiments in the cited paper are all done in vitro and no conclusion can be made about dietary fatty acids and as mentioned in the previous comment the data does not demonstrate direct activation of these PRRs. These sentences have been modified to remove references to direct activation, as the reviewer is correct in pointing out that the data does not definitively support this conclusion.
-
Section 4.1 is long compared to other sections and could be shortened. We agree that Section 4.1 is longer than the other sections; however, given the complex nature of the signaling pathways described, we think the length is required for adequate understanding by a reader not intimately familiar with these pathways.
-
Lines 324-325: The authors state that, “…expression of IRE1α and eIF2α was also drastically reduced”. The expression of eIF2α is significantly reduced but only by ~60%, which is not a drastic reduction. We have changed the wording of this sentence to more accurately reflect the results. The sentence now reads: “In vascular smooth muscle cells (VSMCs), ER stress-induced cell death via TM treatment was significantly increased with loss of NOD2 expression, and correlated with an approximately 60% reduction of IRE1α and eIF2α expression [43].”
Reviewer 3 Report
The review article by Zangara et al., summarize the current available knowledge of how NOD1 and NOD2 may regulate metabolism either by mediating ER stress, calcium signaling or by ROS generation and mitochondrial dysfunction. The authors further highlight the areas of future investigations. Overall, the review is nicely written and provides an up-to date information on the unconventional role of NOD1 and NOD2. I have couple of suggestions to further improve this review:
- Provide a table with the list of NOD1 and NOD2 agonists/antagonists being used/established for bringing metabolic changes with appropriate references.
- Lot of studies have linked role of NOD1 but not NOD2 to obesity and eventually diabetes. Discussing on this aspect will give readers a clear understanding of the differences between NOD1/NOD2. Why NOD1 and NOD2 have opposing effects?
Author Response
We thank the reviewer for taking the time to provide a detailed and constructive review of our manuscript. We have addressed your comments in the manuscript as follows:
- Provide a table with the list of NOD1 and NOD2 agonists/antagonists being used/established for bringing metabolic changes with appropriate references. This is an excellent suggestion and we have added this Table to the manuscript after section 3.
- Lot of studies have linked role of NOD1 but not NOD2 to obesity and eventually diabetes. Discussing on this aspect will give readers a clear understanding of the differences between NOD1/NOD2. Why NOD1 and NOD2 have opposing effects? We agree that the question of why NOD1 and NOD2 have opposing effects is one of the central areas that need further investigation. We attempt to discuss this throughout the manuscript and highlight the differential activation of IRF4 as one potential mechanism that mediates the differential impact of these PRRs.